# Comparison of Microstructure and Mechanical Properties of High Strength and Toughness Ship Plate Steel

**DOI:** 10.3390/ma14195886

**Published:** 2021-10-08

**Authors:** Dong Wang, Peng Zhang, Xingdong Peng, Ling Yan, Guanglong Li

**Affiliations:** 1School of Materials & Metallurgy, University of Science & Technology Liaoning, Anshan 114051, China; chickwong@163.com; 2State Key Laboratory of Metal Material for Marine Equipment & Application, Anshan 114001, China; yanling_1101@126.com (L.Y.); liguanglong@ansteel.com.cn (G.L.)

**Keywords:** ship plate steel, normalizing process, TMCP, EBSD analysis, fracture toughness, fatigue crack growth rate

## Abstract

E36 ship plate steel was, respectively, produced by as rolling and normalizing process (ARNP), and EH36 and FH36 ship plate steel was produced by the thermo-mechanical control process (TMCP) with low carbon and multi-element micro-alloying. The microstructure of the three grades of ship plate steel was composed of ferrite, pearlite, and carbides at room temperature. The average grain size on 1/4 width sections (i.e., longitudinal sections) of the three grades of ship plate steel was, respectively, 5.4 μm, 10.8 μm, and 11.9 μm. EH36 and FH36 ship plate steel had the higher strength due to precipitation and grain boundary strengthening effect, while the E36 ship plate steel had the lower strength due to the recovery phenomenon in the normalizing process. EH36 and FH36 ship plate steel had higher impact toughness due to lower carbon (C) and silicon (Si) content and higher manganese (Mn) content than E36 ship plate steel. E36 ship plate steel had the best plasticity due to the two strong {110} and {111} texture components. The fracture toughness *K_J_*_0.2*BL*(30)_ values of E36 and EH36 and *K_J_*_0.2*BL*_ value of FH36 ship plate steel were, respectively, obtained at 387 MPa·m^1/2^, 464 MPa·m^1/2^ and 443 MPa·m^1/2^. EH36 and FH36 ship plate steel had higher *K_J_*_0.2*BL*(30)_ due to lower C and Si and higher Mn, niobium (Nb), vanadium (V), and aluminum (Al) content than the E36 ship plate steel. The fatigue crack growth rate of E36 ship plate steel was higher than that of EH36 and FH36 ship plate steel due to its higher carbon content and obviously smaller grain size. The analysis results and data may provide a necessary experimental basis for quantitatively establishing the relationship between fracture toughness, yield strength and impact toughness, as well as the relationship between fatigue crack growth rate and both strength and fracture toughness.

## 1. Introduction

It was well known that ship plates were subjected to the strong wind and wave impact and alternating loads in the process of transporting goods. In order to ensure the ship′s safe navigation and reliability, higher strength, toughness, and fatigue performance were required. At the same time, from the perspective of resource conservation and environmental protection, ships need to be lighter, which also requires ship steel plates to have higher strength and toughness. Therefore, ship plate steel with high strength and toughness at the grade of AH, EH, DH or FH has become a research hot-spot [1]. In order to obtain high strength and toughness, low-carbon, high-manganese, composite micro-alloying of Nb, V, titanium (Ti), nickel (Ni) and other elements, TMCP, or the normalizing process after hot rolling have been studied for ship plate steel at higher grades.

For example, the yield strength, tensile strength, and plasticity of DH36 steel plates were improved by adding Mn element [2]. The strength of the DH40 steel plate was improved by adding Nb and Ti elements, whose precipitates played a role in precipitation strengthening and restraining the austenite grain growth [3]. The volume fraction of pearlite in the as-cast structure of the FH40 steel plate was reduced by adding Mg and Zr elements, which induced the nucleation of acicular ferrite and refined ferrite grains. The effect of adding Mg element alone was better than that of adding Zr element alone and Mg-Zr elements [4]. The strength of C-Mn steel after the hot rolling and normalizing process was improved by adding V, V-N, or V-Nb elements to refine the ferrite grains. In addition, excellent strength toughness matching was achieved by adding V-N elements [5].

The equiaxed fine ferrite grain strengthening transformed from non-recrystallized austenite and precipitation strengthening of alloying elements such as Nb, V and Ti element was obtained by controlling the temperature, deformation, and post-rolling cooling regime of TMCP for D36 and E36 ship plates [6]. Compared with the conventional rolling process, the strength of the EH36 steel plate by controlled rolling and TMCP was not significantly improved, but the low-temperature impact toughness was better. EH36 steel plate by TMCP had better comprehensive properties in which the grains were finer [7]. By reducing the rolling temperature by 20 °C and increasing the cooling degree during the ACC process, the TMCP of the DH36 steel plate was smoother and the composition distribution of micro-alloy was more uniform [8]. Compared with TMCP, the normalizing and interstitial quenching processes were economical and effective to improve the properties of steel plates. After the normalizing process, uniform fine ferrite and dispersed pearlite were formed in D36, DH36, DH40, E36 and other steel plates. Although the strength of the steel plates decreased slightly, the plasticity and low-temperature impact properties improved obviously [9,10,11,12,13]. Compared with complete quenching and tempering, the microstructure of EH36 and F550 steel plates after interstitial quenching treatment was ferrite and bainite. The ferrite with low hardness and good plasticity prevented stress concentration and hindered crack growth to improve the low-temperature toughness of the steel plates. The dislocation density in bainite was high, and fine precipitates were precipitated at the dislocation entanglement, which effectively pinned the dislocations in the process of deformation and improved the strength [14,15].

So far, there have been few studies on the comparison of composition, processing technology, and conventional mechanical properties among different grades of ship plate steel with high strength and toughness, the texture composition, fracture toughness and fatigue properties of the steel, and the correlation between the properties and fracture toughness and fatigue crack growth rate. In this study, E36, EH36, and FH36 ship plate steel was selected. Through the comparison and analysis via optical microscope (OM), scanning electron microscope (SEM), energy dispersive spectrum (EDS), electron backscatter diffraction (EBSD), tensile test, fracture toughness, and fatigue crack growth rate test results, the differences of chemical composition, processing technology such as ARNP and TMCP, microstructure, texture, and their effects on overall mechanical properties were determined. Then, the relationships among mechanical properties were established.

## 2. Materials and Methods

### 2.1. Materials

E36, EH36 and FH36 steel ingots were prepared by commercial continuous casting. The melt volume before casting was 8635 kg and the cross-sectional area of a square billet after casting was 250 mm × 250 mm. The designed chemical composition of E36, EH36, and FH36 steel plates was shown in Table 1.

### 2.2. Production Processes Control

E36 steel plate was produced by ARNP and EH36 and FH36 steel plates were produced by TMCP with the billets heated at 1200 °C for 4–5 h. E36 steel plate was only rolled in the austenite recrystallization region, and EH36 and FH36 steel plates were rolled both in the region and the austenite non-recrystallization region. The specific process parameters of rolling, cooling, and heat treatment in the production process of the three grades of ship plate steel are shown in Table 2.

### 2.3. Test Methods

Microstructure test: Specimens of 18 mm × 12 mm × 10 mm were, respectively, cut on 1/4 length, 1/4 width and 1/4 thickness sections of the three grades of steel plate, ground with water sandpapers from 400# to 1000#, polished, and eroded by alcohol solution with 4% concentration nitric acid. The microstructure and EDS were, respectively, tested on the sections at 1/4 length and thickness of the steel plates by ZEISS Vert.A1 OM (Jena, Germany) and JSM-6480lv SEM (Akishima, Tokyo, Japan). EDS test parameters were working voltage of 20 kV, working distance 11 cm and beam spot diameter of 40 nm. The additional microstructure, grain size distribution and texture on the sections at 1/4 width of the steel plates was tested by EBSD system of JSM-6480lv SEM. In order to eliminate the surface stress, the EBSD samples were prepared by mechanical grinding, and then electropolished with 5% perchloric acid alcohol solution at 35 V for 10 s at room temperature.

Performance test: Axial tensile test was carried out by WE-300B universal testing machine (Jinan, China) at room temperature, and the size of specimens was shown in Figure 1. Impact toughness was tested by XJJ-5A Charpy test device (Jinan, China) whose relevant requirements of standard specimens were referred to GB/T 18658-2005. Fracture toughness test was carried out by Instron 8802 hydraulic fatigue testing machine (Norwood, MN, USA) at room temperature. The load stroke of the machine was ±250 kN. A total of ten compact tensile C(T) specimens of each steel plate whose size is shown in Figure 2 were used in the test according to the GB/T 21143-2014 standard [16]. During the test, the COD gauge was used to measure the opening displacement of the specimens at the loading line. The distance of the COD gauge was 5 mm, and the maximum opening displacement was 10 mm. A fatigue crack growth rate test was carried out by Instron 8801 hydraulic fatigue testing machine (Norwood, MN, USA) at room temperature. The load stroke of the machine was ±100 kN. A total of three C(T) specimens of each steel plate whose size was shown in Figure 3 in the test according to the GB/T 6398-2017 standard [17]. The crack growth direction of the specimen was the length direction of the rectangular steel plate. During the test, the crack length was measured by the flexibility method. The COD gauge distance installed on the sample was 5 mm. The maximum opening displacement was 2 mm. The loading mode was axial loading, the stress ratio was 0.03, and the test frequency was 20 Hz. Cracks associated with small plastic zones appeared by step-by-step load reduction. The crack for each level was all prefabricated at constant load control. The reduction range of the adjacent level load was less than 20% and the crack growth length was about 2 mm. Finally, the fatigue crack growth rate *da*/*dN* and stress intensity factor amplitude Δ*K* data of the specimen were obtained by controlling constant load until the specimen fractured. The data points which didn’t meet *W*-*a*
*>* 4/(*K*_fmax_/*R*_p0.2_)^2^ were removed, where W is the specimen width, *a* is crack length; *K*_fmax_ is the maximum stress field intensity factor of prefabricated fatigue crack, *R*_p0.2_ is the yield strength.

## 3. Results and Discussion

### 3.1. Microstructure

Figure 4 showed OM images on 1/4 length sections (i.e., cross sections) of the three grades of steel plate. As can be seen from the images, the microstructure of the three grades of steel plate was all composed of white ferrite and black-and-white pearlite which were distributed at intervals. This was due to the fact that the austenite began to transform into ferrite during the cooling process of normalizing from 822 °C or the controlled cooling process of TMCP from 820 °C and 780 °C. When the temperature drops to the transformation temperature *Ar*_1_, which was, respectively, 685 °C [18], 620 °C, and 577 °C [19] for E36, EH36, and FH36 steel plates, the remaining austenite transformed into pearlite.

Figure 5a–c, respectively, showed the SEM images on 1/4 thickness sections (i.e., parallel to rolling surfaces) of the three grades of steel plate. From the images, it can be found that the microstructure was all composed of a large number of polygonal ferrite (black area), pearlite (P area) and carbide (K area). The pearlite in E36 steel was distributed in strips, which in EH36 and FH36 steel plates were dispersed and fine. The formation mechanism of polygonal ferrite was that the total amount of ferrite was relatively large due to the low C content in the three grades of steel plate, so the ferrite phase continuously formed and thickened along the austenite grain boundaries until many growing ferrite grains contacted each other, thus forming a polygonal structure. Under rolling conditions (See Table 1), the austenite grains of the steel gradually became finer through rolling deformation and recrystallization at the austenite recrystallization stage [20]. In the subsequent cooling process, ferrite was formed at the austenite grain boundary. Due to the low C content in the three grades of steel plates, the total amount of ferrite was high, and a large number of ferrite phases were formed and thickened continuously until they contacted each other, thus forming polygonal ferrite. C element concentration in the austenite increased at the austenite-ferrite phase boundary, resulting in the precipitation of carbides at the phase boundary due to the formation of ferrite. The carbides precipitated from the austenite can only be distributed in fine particles, since the amount of C atoms provided by the unit volume austenite was very small. EDS in Figure 5d–f showed that the Mn content of ferrite phase in the three grades of steel plate was 1.47 wt%, 2.1 wt% and 1.87 wt%, in turn.

In order to examine the microstructure characteristics of the steel plates, EBSD analysis was carried out. Figure 6 showed EBSD grain images and grain size distribution of 1/4 width sections (i.e., longitudinal section) of the three grades of steel plate, which showed the morphology of ferrite grains more clearly than OM and SEM images. According to the grains’ morphology, it can be judged that all three grades of steel plate had recrystallized, but the degree of recrystallization was different. The degree of recrystallization of the EH36 and FH36 steel plates was higher, while that of the E36 steel plate was lower. This was due to the fact that the EH36 and FH36 steel plates had been rolled in the austenite unrecrystallized zone, resulting in large strain accumulation and strain storage energy in austenite, which not only promoted the transformation from austenite to ferrite, but also made the nucleation rate and growth rate of ferrite larger during dynamic recrystallization, so the recrystallization can be completed quickly. Since the E36 steel plate didn’t undergo rolling in the unrecrystallized austenite zone, a large strain wasn’t accumulated in the austenite, and the strain storage energy was small. Therefore, the ferrite recrystallization progressed slowly and the degree of recrystallization was low. It was worth noting that many newly formed micro-grains can be observed in the E36 steel plate at the boundary of coarse parent grains, especially at the intersection of three grain boundaries such as T area in Figure 6a (that is, the intersection was the preferred nucleation position). According to F. Fisher′s paper, this was due to the fact that when the grain boundary sliding was the main deformation mode, the sliding along the grain boundary ended at the three-phase point, resulting in stress concentration and deformation superposition at the grain boundary and the formation of recrystallized nuclei, accompanied by the migration of grain boundaries to adapt to new grains [21]. The average grain size of three grades of steel plates was, respectively, 5.4 μm, 10.8 μm and 11.9 μm (i.e., E36 < EH36 < FH36). The average grain size of E36 steel plate was obviously less than that of EH36 and FH36 steel plates, mainly due to the Mn and phosphorus (P) content in the E36 steel plate (See Table 2) which promoted the austenite grain growth were lower and the austenite grains didn’t grow obviously, the austenite and ferrite grains transformed by austenite were fined. As the cooling rate of EH36 steel plate was faster than that of FH36 steel plate, the grain size of EH36 steel plate was smaller.

### 3.2. Mechanical Properties

Figure 7, respectively, showed the mechanical properties of the E36, EH36, and FH36 steel plates. As can be seen from Figure 7a, the yield strength and tensile strength values of the three grades of steel plate were all higher than 355 MPa and 490 MPa as required by the classification society rule. Their magnitude relationship of strength values was as follows: EH36 > FH36 > E36. This was due to the fact that a large amount of deformation dislocations were formed and retained in the austenite phase during the controlled rolling process of EH36 and FH36 steel plates. In the process of rapid cooling after rolling, fine Nb, V, and Ti carbides precipitated from austenite (See Figure 5a–c). On the one hand, these carbides promoted ferrite nucleation and further refinement of ferrite grains. On the other hand, they pinned dislocations and produced significant dislocation strengthening effect, making EH36 and FH36 steel plates have a higher strength than the E36 steel plate.

In contrast, the strength of the E36 steel plate was lower. This was since, although the strength increased during the normalizing process due to the refinement of grains, the C atoms continued to diffuse and redistributed in the supersaturated solid solution α-Fe, which significantly reduced the super saturation of C atoms in the ferrite. The recovery phenomenon occurred in the normalizing process. Some dislocations tend to be linearized and flattened from tangled random distribution, and the dislocation density decreased, resulting in the aggregation effect of solute atoms on dislocations weakening. Due to the comprehensive effect of the above causes, the strength decreased, but the plasticity (See Figure 7b) increased, which was higher than that of EH36 and FH36 steel plates. The strength of the EH36 steel plate was higher than that of the FH36 steel plate due to higher C and Mn contents, but the elongation of the former was lower than the latter.

As can be seen from Figure 7c, the magnitude relationship of impact toughness values of the three grades of steel plate was as follows: EH36 > FH36 > E36. It was well known that carbon deteriorates the toughness of steel. Therefore, as long as there was no problem in strength, low carbon content was beneficial to the improvement of toughness. Silicon can improve the yield strength, but can also lead to the deterioration of toughness. On the contrary, Mn can improve toughness. Especially when the Mn/C content ratio was more than 3, such as the Mn/C ratio in the three steel plates in this study, the toughness of the steel greatly improved due to the presence of the Mn element [22]. The impact toughness value of the E36 steel plate was significantly lower than that of the EH36 and FH36 steel plates, since C and Si content in the former were higher than that in the latter and the Mn content (See Figure 5d–f) in the former was significantly lower than that in the latter. The impact toughness value of the EH36 steel plate was higher than that of FH36 steel plate, which was mainly related to the higher Mn content in EH36 steel plate and the lower P and chromium (Cr) content that reduced impact toughness.

### 3.3. Strengthening Mechanism

In recent years, the strengthening mechanism derived from the study of mechanical properties of steel can be divided into four types. These were solid solution strengthening, grain boundary strengthening, dislocation strengthening, and precipitation strengthening. The strength of steel plates depended on the sum of the strength increment caused by these four strengthening mechanisms (See Formula (1)) [23].
*σ_ys_* = *σ_ss_* + *σ_gb_* + *σ_dis_* + *σ_pre_*(1)
where, *σ_ys_* is the yield strength, MPa; *σ_ss_*, *σ_gb_*, *σ_dis_* and *σ_pre_* are, respectively, the strength increment caused by solid solution strengthening, grain boundary strengthening, dislocation strengthening, and precipitation strengthening (MPa).

#### 3.3.1. Solid Solution Strengthening

Solid solution strengthening can improve the strength of steel by changing the composition of the material. The principle of solid solution strengthening was that the solute atoms in steel can block the movement of dislocations and reinforce the properties of steel. The degree of solid solution strengthening was related to the number of solute atoms. In the low-carbon steel, the increase of solid solution strengthening and the amount of solid solution atoms can be approximately regarded as a linear relationship, which can be calculated by Formula (2):*σ_ss_* = ∑*K_i_*[*M_i_*](2)
$$\sigma_{s}=\sum K_{i}[M_{i}]$$
where, *K_i_* is the yield strength increment per 1% mass fraction of solid solution element (See Table 3 [23,24]), MPa; [*M_i_*] is the mass percentage of the element, %.

By substituting the data in Table 1 and Table 3 into Formula (2) at the end, the *σ_ss_* value of E36, EH36 and FH36 steel plates was respectively calculated to be 126.2 MPa, 117.51 MPa, and 97.72 MPa.

#### 3.3.2. Grain Boundary Strengthening

Since the grain boundaries of the structured grains in the steel blocked the movement of dislocations, it had a certain strengthening effect on the strength of the steel (that is, the grain boundary strengthening effect affected the yield strength of the material). In general, low-carbon steel, the influence of the strengthening effect on the strength was proportional to the negative one-half of the grain size in the steel (that is, the Hall-Petch Formula (3)) [25]:*σ_gb_* = *k_y_d^−^*^1/2^ + *σ*_0_(3)
where, *k_y_* is 17.4 MPa·mm^1/2^ for low carbon steel; *d* is the average diameter of ferrite grains, mm; and *σ*_0_ is the lattice friction shear stress, 53 MPa.

By substituting the average grain size of the three grades of steel plate obtained by EBSD into Formula (3) at the end, the *σ_gb_* value of the E36, EH36, and FH36 steel plates was, respectively, 236.78 MPa, 167.43 MPa, and 159.51 MPa.

#### 3.3.3. Dislocation Strengthening

The interaction between the original dislocations in the steel and the newly generated dislocations in the stress process and the resistance of the dislocations to the movement of the newly generated dislocations leads to dislocation strengthening. The strength increment caused by dislocation strengthening was proportional to one-half of the dislocation density in the material. The *σ_dis_* value can be calculated by the Formula (4) [26]:*σ_dis_* = *αMGbρ*^1/2^(4)
where, *M* is the Taylor orientation factor with a value of 2.75 [27]; *α* is a numerical factor, 0.435 [28]; *G* is the shear modulus, 80650 MPa; *b* is the Burgers vector, 0.248 nm; and *ρ* is the dislocation density of the material, nm^−2^. Assuming that the dislocation density of E36 steel plate after normalizing treatment was 10^13^ m^−2^ [29] and the dislocation density of ferrite formed by austenite transformation of EH36 and FH36 steel plates was 5 × 10^13^ m^−2^ [30], which were substituted into the Formula (4), the *σ_dis_* value of E36, EH36, and FH36 steel plates was, respectively, 75.66 MPa, 169.19 MPa, and 169.19 MPa.

#### 3.3.4. Precipitation Strengthening

When a material undergoes plastic deformation under stress, there were many ways of interaction between the second phase particles and dislocations (mainly as a cutting mechanism and a bypass mechanism, which form resistance to the movement of dislocations and produce precipitation strengthening effect) [26,31]. J. Fu et al. considered that the original dislocation density was related to the pinning effect of the precipitated second phase particles. After calculating the strength increment caused by precipitation strengthening of the second phase particles, the resistance of the original dislocation density to the newly generated dislocation movement can no longer be considered that was, the *σ_dis_* value was no need to consider the calculation [32]. So, under certain conditions, the *σ_ys_* value was equal to the sum of *σ_ss_*, *σ_gb_* and *σ_pre_* values.

Therefore, from the *σ_ys_* values minus *σ_ss_* and *σ_gb_*, the *σ_pre_* values can be calculated in E36, EH36 and FH36 steel plates which were, respectively, 40.02 MPa, 172.06 MPa, and 198.77 MPa. FH36 steel plate contained more carbide-forming elements Mn, Nb, V, Ti, and Cr, so the precipitated Cr_23_C_6_, Mn_3_C, VC, TiC and NbC carbides [24] and the rich copper (Cu) phase precipitated by Ni were correspondingly more [33] and the *σ_pre_* value was the highest. For E36 steel plate, it was difficult to form a local supersaturated solid solution zone in the slow cooling process, and the number of precipitated phases should be least, so the *σ_pre_* value was the lowest.

The results show that the strength increment caused by precipitation strengthening of EH36 and FH36 steel plates was the highest. The higher the grain boundary strengthening, the lower the solid solution strengthening. And for E36 steel plate, the highest was the grain boundary strengthening, and the higher the solid solution strengthening, the lower the precipitation strengthening.

### 3.4. Texture Components

Figure 8 showed the measured constant φ_2_ = 45° ODF sections for the E36, EH36 and FH36 steel plates. Figure 9 showed the ideal texture components on the φ_2_ = 45° ODF section of the bcc crystal and the main texture components of the three steel plates in Figure 8. It can be seen from Figure 9 that the Euler angles of the main texture components in the three grades of steel plates were close to the Euler angles of the adjacent ideal texture components, which can be approximately classified as the adjacent ideal texture components. The Euler angles corresponding to the texture components that cannot be classified in Figure 9 were converted into Miller indexes by Textools software. The Miller indexes corresponding to the texture components in the three grades of steel plates were shown in Table 4. It can be seen from the table that all three grades of steel plates contained rolling texture component {111}<uvw> or {hkl}<110> from bcc metals [34].

The references of IF steel and bake hardening steel with very low carbon content can be used for reference since there were few references on texture evolution of the three grades of steel plate. The formation of {111}<uvw> may be related to the rolling deformation mechanism in the austenite region and the orientation relationship when austenite transformed to ferrite. The hot rolling deformation in the austenite region was mainly realized by cross slip of screw dislocations and climbing of edge dislocations, in which the cross slip of screw dislocations was determined by the start of {111}<110> slip systems. The dislocations climbed easier at the higher rolling temperature. Thus, the alternating start of the slip system may result in the formation of weak {110} texture components in the final hot deformed austenite, and {111} texture appeared when the austenite transformed into ferrite [35]. Y. Park et al. reported that when titanium-containing IF steel was rolled in the α-ferrite region, {112}<110> texture component developed [36]. In addition, three grades of steel plate also contained some other texture components, such as the common recrystallized $\left(334)[4\bar{8}3\right]$ texture component in bcc crystals.

It was noteworthy that for E36 Steel, due to its more Si content, there was a unique normalizing $\left(001)[0\bar{1}0\right]$ texture. In addition, the texture of the E36 steel plate contains {110}<110> texture component, which usually comes from the deformation zone. This indicated that the E36 steel plate didn’t have completely recrystallized, which was consistent with the grain morphology of the E36 steel plate in Figure 6a [37]. For the EH36 steel plate, the orientation distribution function values of the three texture components were very small, so the texture strength of EH36 steel was obviously weaker than that of E36 and FH36 steel plate. For the EH36 steel plate, due to its large reduction ratio, the <112> texture was weak, which was in agreement with the characteristic of weak texture obtained by the traditional hot rolling process [38]. The favorable texture components were {110}, {111} and {112} texture components for improving plasticity [39]. Thus, the E36 steel plate had the best plasticity due to two strong {110} and {111} texture components. FH36 contained only one strong {111} texture component, so the plasticity was slightly lower, while EH36 has the lowest plasticity due to the two weak {111} and {112} texture components.

### 3.5. Fracture Toughness

Figure 10 showed the *J*-Δ*a* resistance curve (red line) of the FH36 steel plate obtained according to the measurement data by fitting Formula (5) provided by the ISO 12135:2016 standard.
*J* = *α* + *β* (Δ*a*)*^γ^*(5)
where, *J* is fracture toughness, kJ/m^2^; Δ*a* is crack growth length, mm; and *α*, *β* and *γ* are fitting coefficients.

After fitting, the parameters in Formula (1) were as follows: *α* = 0, *β* = 1128, *γ* = 0.57 and adjusted determination coefficient *R*^2^ = 0.77. Thus, the fitting formula (See Formula (6)) was obtained as the following:*J* = 1128 (Δ*a*)^0.57^(6)

The formula was *J* = 2074Δ*a* corresponding to passivation line in Figure 10, and the *J*-Δ*a* resistance curve intersected the passivation line which was shifted 0.2 mm to the right. By the location of the intersection, we can obtain the *J*_0.2*BL*__(50)_ value at 846 kJ/m^2^. Then, the following seven criteria can be used to determine whether the *J*_0.2*BL*__(50)_ was the *J*_0.2*BL*_ which was insensitive to specimen size. If all seven criteria were met, the *J*_0.2*BL*__(50)_ was the *J*_0.2*BL*_. Otherwise, only the *J*_0.2*BL*__(50)_ could be obtained, which was sensitive to specimen size. The seven criteria are as follows:

i. At least one point (actual three points) between 0.1 mm and 0.3 mm shifted passivation line; at least two points (actual five points) between 0.1 mm and 0.5 mm shifted the line, and at least six points (actual nine points) were in the valid range.

ii. *α* ≥ 0, *β* ≥ 0, 0 ≤ *γ* ≤ 1 (actual *α* = 0, *β* = 1128, *γ* = 0.57)

iii. (*dJ*/*d*Δ*a*)_0.2*BL*_ < 1.875*R_m_* (*R_m_* was tensile strength, 553 MPa, actual (*dJ*/*d*Δ*a*)_0.2*BL*_ = 799, 1.875*R_m_* = 1037)

iv. *a*_0_ > 40*J*_0.2*BL*_/(*R_p_*_0.2_ + *R_m_*) (*a*_0_ was the average length of pre-crack, mm), *B* > 40*J*_0.2*BL*_/(*R_p_*_0.2_ + *R_m_*), (*W*-*a*_0_) > 40*J*_0.2*BL*_/(*R_p_*_0.2_ + *R_m_*) (*B* was specimen thickness, mm, *R_p_*_0.2_ was yield strength, 456 MPa; *W* was specimen width, mm, actual *a*_0_ = 53.34, *B* = 50.07, *W*-*a*_0_ = 46.58, 40*J*_0.2*BL*_/(*R_p_*_0.2_ + *R_m_*) = 33.54)

v. *a*_0*i*_-*a*_0_ < 0.1*a*_0_, *a_i_*-*a* < 0.1*a* (*I* = 2~8)(*a*_0*i*_ was the crack length at the position *i* of the leading edge of the pre-crack, mm, *a* was crack length, mm)

vi. 0.45*W* < *a*_0_ < 0.70*W*, the pre-crack length exceeded the larger of 1.3 mm and 2.5%*W* (actual *a*_0_ = 0.534*W* and the pre-crack length was about 5–8 mm)

vii. 0.5*W* ≤ Δ*a*_max_ ≤ 0.1(*W*-*a*_0_) (actual Δ*a*_max_ = 1.676)

Since the above seven criteria were all met, the *J*_0.2*BL*_ value of the FH36 steel plate can be obtained at 846 kJ/m^2^, which was insensitive to the specimen size. Furthermore, its fracture toughness *K_J_*_0.2*BL*_ value can be calculated as 443 MPa·m^1/2^ by conversion Formula (7) [40]).
*K* = [*E*/(1 − *μ*^2^)*J*]^1/2^(7)
where, *K* is the fracture toughness, MPa·m^1/2^; *E* is the elastic modulus, 207GPa; and *μ* is Poisson ratio, 0.33.

Figure 11 and Figure 12 separately showed the *J*-Δ*a* resistance curves of E36 and EH36 steel plates and their *J*_0.2*BL*(30)_ values at 644 kJ/m^2^ and 926 kJ/m^2^. Their fracture toughness *K_J_*_0.2*BL*(30)_ values only can be obtained at 387 MPa·m^1/2^ and 464 MPa·m^1/2^, respectively, since all the seven criteria were unsatisfied. Hence, the magnitude relationship of their fracture toughness *K_J_*_0.2*BL*(30)_ values was as follows: EH36 > FH36 > E36.

The fracture toughness *K_J_*_0.2*BL*(30)_ of EH36 and FH36 steel plates was significantly higher than that of E36 steel plates, which can be regarded as steel plates with higher toughness that was, it was uneasy to occur low stress brittle fracture. This was related to its lower C and Si content and higher content of Mn, Nb, V, and Al, which were beneficial to fracture toughness (See Table 1). Although the C and Mn content in EH36 steel plate was higher than that of FH36 steel plate, the content of sulfur (S), P, Cu, and Cr which was unfavorable to the fracture toughness was lower. In general, the fracture toughness of the EH36 steel plate was higher than that of the FH36 steel plate. This was consistent with the general law of material toughness that the higher the impact toughness, the higher the fracture toughness (See Figure 7c) [41].

### 3.6. Fatigue Cracks Growth Rate

Figure 13 showed the *da/dN-*Δ*K* data points and corresponding fitting lines of the three grades of steel plate. From the figure, the data points showed an approximate linear relationship (that is, their crack growth process followed the Paris Formula (8)) [42]:*da*/*dN* = *C* (Δ*K*)*^m^*(8)
where, *a* is crack length, mm; *N* is the corresponding number of load cycles, cycle; *da*/*dN* is fatigue crack growth rate, mm/cycle; Δ*K* is stress intensity factor amplitude, MPa·m^1/2^; and *C* and *m* are Paris material constants.

Thus, the parameters in the Paris formula were obtained by double logarithmic linear fitting of data points from E36, EH36 and FH36 steel plates (See Table 5) and the correction determination coefficients *R*^2^ was, respectively, 0.958, 0.963, and 0.955, which showed that the fitting lines were in good agreement with the data points.

As can be seen from Table 5, the *m* value in the Paris formula of the three grades of steel plate changed in a very narrow range from three to four (that is, the crack growth curve was relatively flat, and the crack growth rate was slower, which showed that high-strength steel had better fatigue properties to a certain extent). At the same time, it can be seen that the higher the yield strength of the three grades of steel plate (EH36 > FH36 > E36), the lower the fatigue crack growth rate parameter m (EH36 < FH36 < E36), and the flatter the crack growth rate curve, indicating that the slower the crack growth rate increased, the stronger the steel’s ability to resist fatigue crack growth (that is, the fatigue performance becomes better with the increase of strength) [43]. As for the fatigue crack growth mechanism, there was a plastic passivation model (C. Laid and G.C. Smith Model), which indicated that the crack tip region of the material can expand forward for a certain length by repeated plastic deformation under the action of shear stress (i.e., completing a change process of shape from sharpening to passivation to sharpening in each cycle) [44]. It can be seen from this model that increasing the yield strength of materials was an important method to prevent and slow down the growth of fatigue cracks.

According to the formulae in Table 5, when the Δ*K* value exceeded 21 MPa, the magnitude relationship between the fatigue crack growth rate values of the three grades of steel plate was: EH36 < FH36 < E36, which was opposite to the relationship between their fracture toughness values. This was also consistent with the law that the higher the fracture toughness *K_J_*_0.2*BL*(30)_, the lower the fatigue crack growth rate *da/dN* [45]. As a rule, with the increase of carbon concentration, the toughness decreased, the sensitivity to crack increased, and the crack growth rate increased under the same intensity factor. As the grain size decreased, the fatigue crack growth threshold decreased, and the fatigue crack growth rate also increased [46]. Although the C content of the EH36 steel plate was higher than that of the FH36 steel plate, and the grain size of the EH36 steel plate was slightly smaller than that of FH36 steel plate, leading to the increase of fatigue crack growth rate, the Mn content in the EH36 steel plate (which was beneficial to the fracture toughness) was higher, and the S, P, Cu and Cr content in the EH36 steel plate which was unbeneficial to fracture toughness was lower, resulting in decrease of the fatigue crack growth rate. As a result, the two *da*/*dN*-Δ*K* fitting lines of EH36 and FH36 steel plates intersected when the Δ*K* value was equal to 21 MPa. When the Δ*K* value exceeded 21 MPa, the fatigue crack growth rate of EH36 steel plate was slightly lower. By comparing the effect of yield strength and fracture toughness on the fatigue crack growth rate of the three grades of steel plate, it can be understood that the effect of fracture toughness was more significant than that of yield strength [47].

## 4. Conclusion

In this study, E36 ship plate steel produced by the ARNP process, and EH36 and FH36 ship plate steel produced by TMCP was trial produced in an industrial environment. Through the comparison of microstructure and mechanical properties, we could draw the following conclusions:The microstructure of E36, EH36, and FH36 ship plate steel at room temperature was composed of polygonal ferrite, pearlite, and granular carbides. The average grain sizes on 1/4 width sections (i.e., longitudinal sections) of the three grades of ship plate steel were, respectively, 5.4 μm, 10.8 μm and 11.9 μm. E36 ship plate steel has the lowest Mn and P content, so the average grain size was the least.EH36 and FH36 ship plate steel had the higher strength due to precipitation strengthening and grain boundary strengthening, while E36 ship plate steel had the lower strength due to the recovery phenomenon in the normalizing process. E36 ship plate steel had the best plasticity due to the strong {110} and {111} texture components, and obviously lowest impact toughness due to the higher C and S contents and lower Mn content.EH36 and FH36 ship plate steel had lower C and Si contents, higher Mn, Nb, V and Al contents resulting in higher *K_J_*_0.2*BL*(30)_ than E36 ship plate steel. The *K_J_*_0.2*BL*(30)_ of EH36 ship plate steel was higher than FH36 ship plate steel, which was related to the relatively higher Mn content and lower S, P, Cu, and Cr contents.When the Δ*K* value exceeded 21 MPa, E36 ship plate steel had a relatively larger fatigue crack growth rate due to the higher C content and significantly smaller grain size than EH36 and FH36 ship plate steel.The tensile test can be used as a simple, cheap, and stable test to evaluate impact toughness, fracture toughness and fatigue crack growth rate of E36, EH36, and FH36 ship plate steel.

## Figures and Tables

**Figure 1 materials-14-05886-f001:**
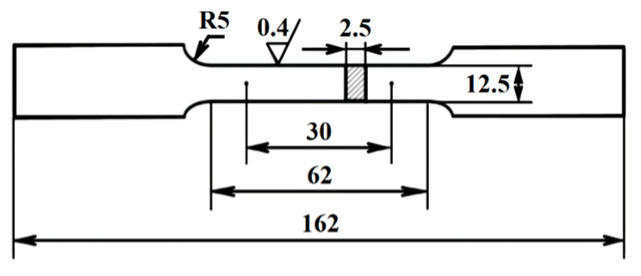
Size of specimen in the tensile test.

**Figure 2 materials-14-05886-f002:**
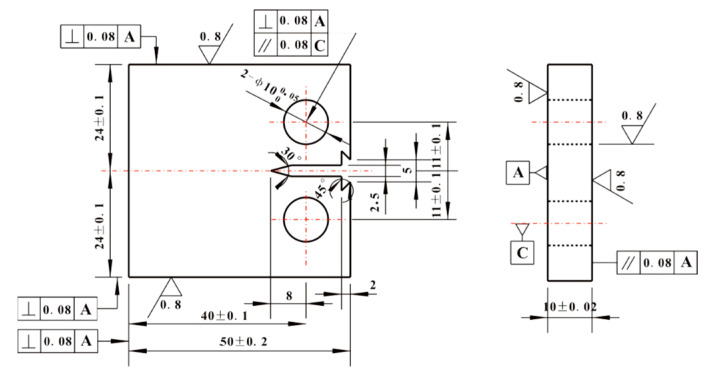
Size of C(T) specimen in fracture toughness test.

**Figure 3 materials-14-05886-f003:**
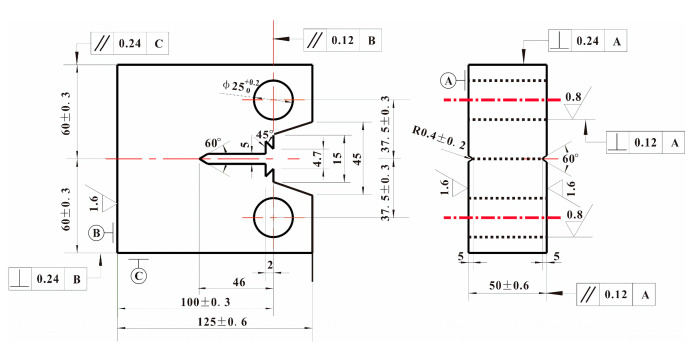
Size of C(T) specimen in fatigue crack growth rate test.

**Figure 4 materials-14-05886-f004:**

OM images on 1/4 length sections of E36 (**a**), EH36 (**b**) and FH36 (**c**) steel plates.

**Figure 5 materials-14-05886-f005:**
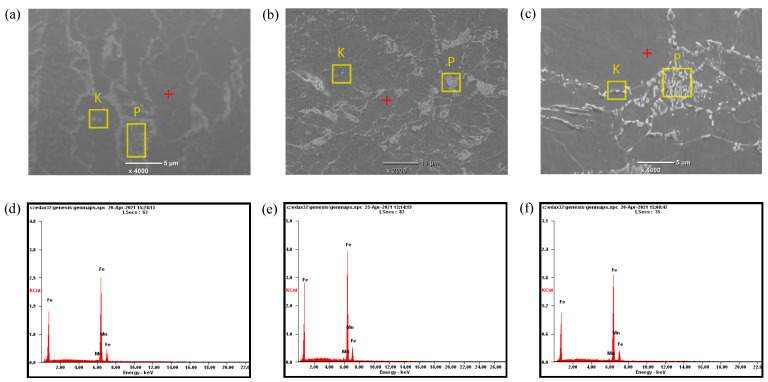
SEM images and EDS on 1/4 thickness sections of E36 (**a**,**d**), EH36 (**b**,**e**) and FH36 (**c**,**f**) steel plates.

**Figure 6 materials-14-05886-f006:**
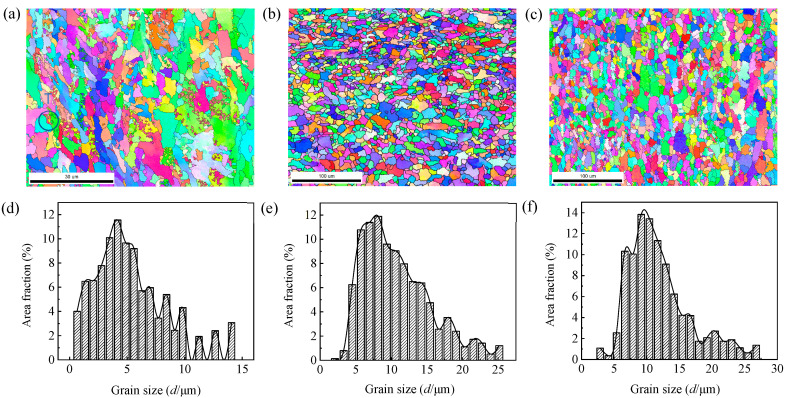
EBSD grain images on 1/4 width sections and grain size distribution of E36 (**a**,**d**), EH36 (**b**,**e**), and FH36 (**c**,**f**).

**Figure 7 materials-14-05886-f007:**
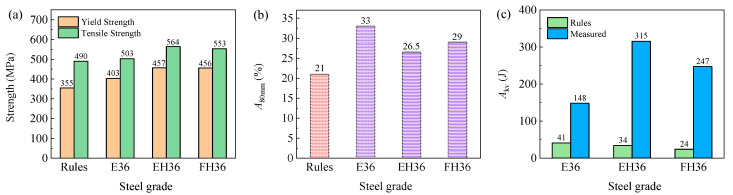
Strength (**a**), elongation (**b**), and impact toughness (**c**) indexes of the three grades of steel plate.

**Figure 8 materials-14-05886-f008:**
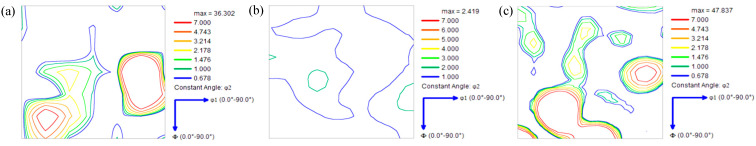
Constant *φ*_2_ = 45° ODF sections of E36 (**a**), EH36 (**b**) and FH36 (**c**) steel plates.

**Figure 9 materials-14-05886-f009:**
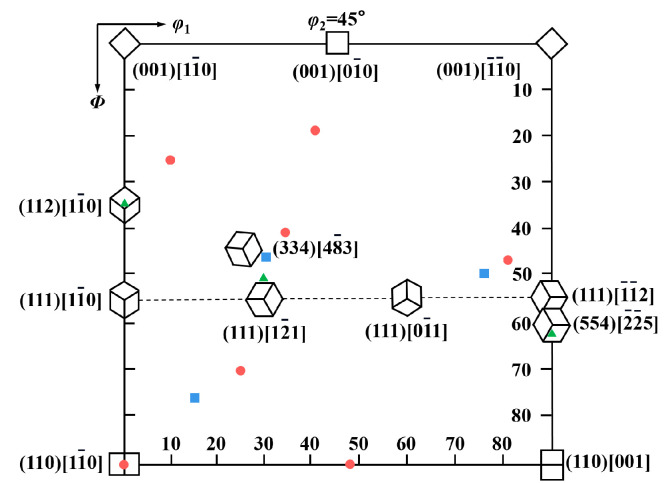
Constant *φ*_2_ = 45° ODF section of bcc crystal and the texture of E36 (**●**), EH36 (**▲**) and FH36 (**■**) steel plates.

**Figure 10 materials-14-05886-f010:**
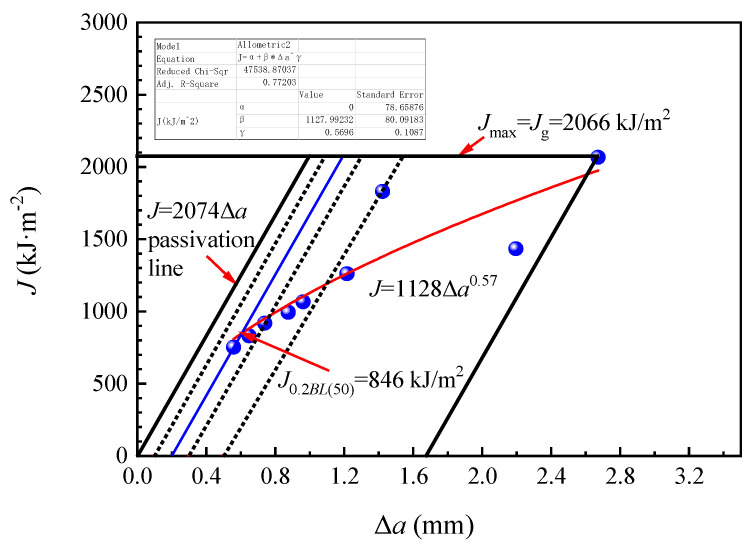
*J*-Δ*a* resistance curve of FH36 steel plate.

**Figure 11 materials-14-05886-f011:**
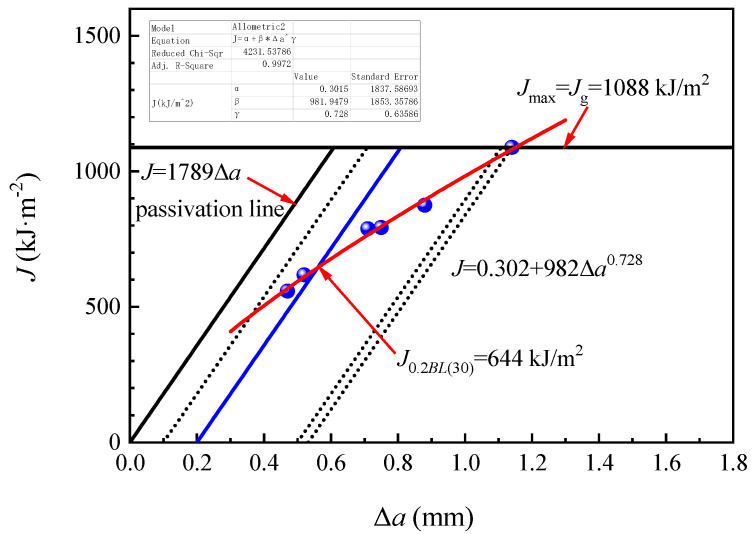
*J*-Δ*a* resistance curve of E36 steel plate.

**Figure 12 materials-14-05886-f012:**
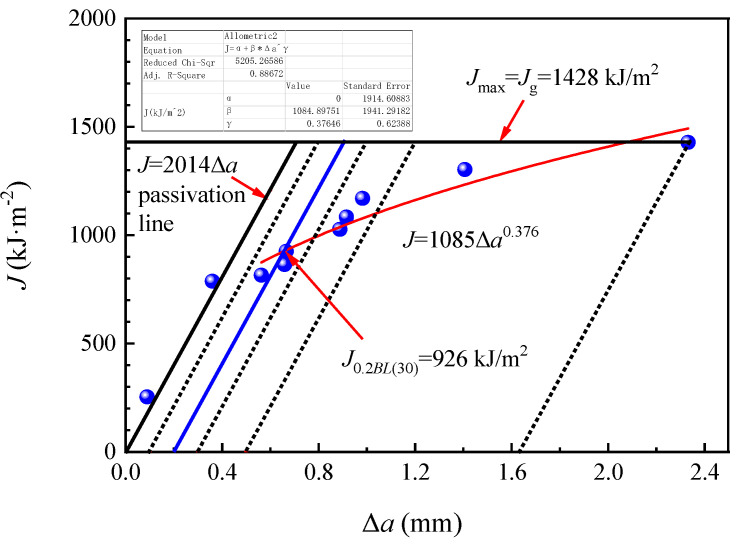
*J*-Δ*a* resistance curve of EH36 steel plate.

**Figure 13 materials-14-05886-f013:**
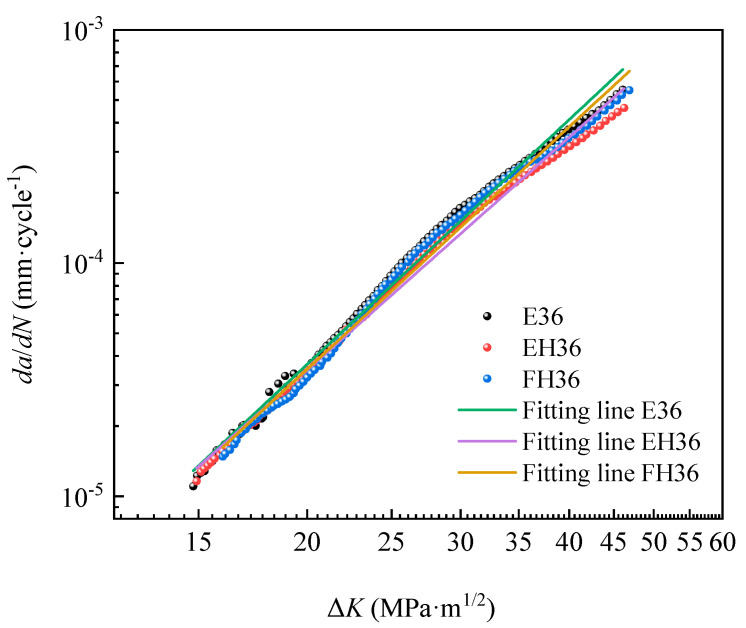
*da*/*dN*-Δ*K* data points and fitting lines of the three grades of steel plate.

**Table 1 materials-14-05886-t001:** Chemical composition of the three grades of ship plate steel (mass fraction, %).

Grade	Chemical Composition												
	C	Si	Mn	S	P	Nb	V	Ti	Al	Cu	Cr	Ni	Mo
E36	0.14	0.3	1.25	0.0021	0.015	0.026	0.002	0.032	0.032	0.016	0.05	0.042	0.01
EH36	0.10	0.15	1.55	0.003	0.018	0.04	0.05	0.012	0.035	-	-	-	-
FH36	0.06	0.15	1.35	0.005	0.022	0.04	0.045	0.017	0.045	0.12	0.18	0.38	-

**Table 2 materials-14-05886-t002:** Rolling, cooling, and heat treatment processes parameters of the three grades of ship plate steel.

Grade	Rolling Process Parameters	Cooling Process Parameters	Heat Treatment Parameters
E36	Hot rolling to 110 mm, final rolling temperature 822 °C, final thickness 50 mm	Air cooling	Heating at 860 °C for 90 min and air cooling
EH36	At first stage rough rolling to 130 mm, at second stage start rolling temperature 860 °C in the of finish rolling, final rolling temperature 830 °C, rolling deformation rate 2–5/s, final thickness of steel plate 50 mm, total reduction rate 80%	Open cooling temperature 820 °C, the final cooling temperature 560 °C, and the cooling rate 10.4 °C/s	-
FH36	At first stage rough rolling to 150 mm, at second stage start rolling temperature 830 °C of finish rolling, final rolling temperature 800 °C, rolling deformation rate 2–4/s, final thickness of steel plate 60 mm, total reduction rate 76%	Open cooling temperature 780 °C, the final cooling temperature 500 °C, and the cooling rate 7.3 °C/s	-

**Table 3 materials-14-05886-t003:** Yield strength increment *K_i_* of solid solution elements per 1% mass fraction in ferrite, MPa.

Element	C	Si	Mn	Al	Cu	Ni	Cr	V	Ti	P
*K_i_*	360	83	37	60	38	0	−30	3	80	470

**Table 4 materials-14-05886-t004:** Main texture components in the three grades of steel plate.

Grade	Strong Texture Component	Weak Texture Component
E36	$\left(111)[1\bar{2}1\right]$, $\left(110)[2\bar{2}3\right]$, $\left(111)[\bar{1}\bar{1}2\right]$, $\left(110)[1\bar{1}0\right]$	$\left(112)[1\bar{1}0\right]$, $\left(334)[4\bar{8}3\right]$, $\left(001)[0\bar{1}0\right]$
EH36	-	$\left(112)[1\bar{1}0\right]$, $\left(111)[1\bar{2}1\right]$, $\left(554)[\bar{2}\bar{2}5\right]$
FH36	$\left(111)[\bar{1}\bar{2}3\right]$, $\left(331)[2\bar{3}1\right]$	$\left(334)[4\bar{8}3\right]$

**Table 5 materials-14-05886-t005:** Paris formulae for three grades of steel plate.

Grade	Paris Formula
E36	*da*/*dN* = 1.07 × 10^−9^ (Δ*K*)^3.49^
EH36	*da*/*dN* = 1.68 × 10^−9^ (Δ*K*)^3.32^
FH36	*da*/*dN* = 1.13 × 10^−9^ (Δ*K*)^3.45^

## Data Availability

Not applicable.

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
