# Peer review of "Comparison of Microstructure and Mechanical Properties of High Strength and Toughness Ship Plate Steel"

_materials, 2021, doi:10.3390/ma14195886_

Round 1

Reviewer 1 Report

The significance of this research is extremely pertinent. Although much has been published regarding microalloying in these plate grades, I like the author's focus on the lower sulfur and phosphorous levels. Certainly the grain size is important, but often compromises are made with the residual elements and sulfur and phosphorous offsetting the benefits of finer grain size.

The inclusion of not just impact strength, but also fatigue and fracture toughness is often not reported . It is very good to connect all three mechanical property results. Certainly, from both a metallurgical and design perspective, these properties are of paramount importance.

The sound conclusion of the sulfur and phosphorous effect on mechanical properties is soundly discussed and concluded. The introduction of lower manganese in the E36 grade is useful information for the steel community.

Author Response

Point 1: English language and style are fine/minor spell check required. 

Response 1: Minor spelling and grammar have been checked. Please see the attachment.

Thank you so much for your positive comments on our paper. Best wishes to you.

Reviewer 2 Report

shipbuilding steels require development, so the topic is relevant

literature review can be improved, this class of steels was broadly studied in the past

results contain many sections, this is ok, but it would be better to present less in volume and more in depth analysis

microstructural characterisation could be improved, and, because of this, the analysis of properties development mechanisms is lacking evidence

for additional comments please see the pdf file

Author Response

Thank you a lot for your review and comments for our manuscript. The comments are so helpful and important for us.
In this revised paper, all contents have been greatly added or revised, such as Sections 3.2, 3.3, 3.4 and 3.6. The English language, style and grammar has been improved. By adding a large number of relevant references, we think the references are sufficient. 
Please see the attachment. Thank you so much for your review again and best wishes to you.

Reviewer 3 Report

Two important remarks:

  • extend introduction, for readers is important to add what is the topic of the paper,
  • add more references, mostly review paper

foe example:

Kvackaj T, Bidulská J, Bidulský R. Overview of HSS Steel Grades Development and Study of Reheating Condition Effects on Austenite Grain Size Changes. Materials. 2021; 14(8):1988. https://doi.org/10.3390/ma14081988

Heibel, S. Damage mechanisms and mechanical properties of high-strength multiphase steels Materials Volume 11, Issue 59 May 2018 Article number 761 DOI 10.3390/ma11050761

Martínez, C., Briones, F., Villarroel, M., Vera, R. Effect of atmospheric corrosion on the mechanical properties of SAE 1020 structural steel       2018 Materials 11(4), 591

Branco, R., Berto, F. Mechanical behavior of high-strength, low-alloy steels 2018 Metals 8(8),610

Kozłowska, A., Grzegorczyk, B., Morawiec, M., Grajcar, A. Explanation of the PLC effect in advanced high-strength medium-mn steels. A review 2019 Materials 12(24),4175

Wang, Y., Zhu, L., Zhang, Q., Zhang, C., Wang, S. Effect of Mg treatment on refining the microstructure and improving the toughness of the heat-affected zone in shipbuilding steel 2018 Metals 8(8),616

Author Response

Thank you a lot for your review and comments for the manuscript. The comments are so helpful and important for us.
In this revised paper, all contents have been greatly added or revised, such as Sections 3.2, 3.3, 3.4 and 3.6. The English language, style and grammar has been improved. By adding a large number of relevant references, we think the references are sufficient. 
Please see the attachment. Thank you so much for your review again and best wishes to you.

Reviewer 4 Report

In my opinion, this paper appears to be a preliminary rather than a final and conclusive analysis. The introduction requires more depth and in my opinion does not provide sufficient background. More relevant references must be added to this introduction. The structure of the paper must be reorganized and the objectives of the study must be detailed more clear. The discussion of the results requires more depth and interpretation of the results is needed. Extensive editing of English language and style is required by a native English speaking colleague.

Author Response

(The authors gave the same response as above.)

Round 2

Reviewer 2 Report

the authors have answered all the comments and the paper can be published now

Reviewer 4 Report

In my opinion manuscript has been improved and I suggest publication of the manuscript